# Cooking with the Seasons for Health (CwS4H): An Innovative Intervention That Links Nutrition Education, Cooking Skills, and Locally Grown Produce to Increase Vegetable Intake among Limited-Resource Parent–Child Dyads in Rural Washington

**DOI:** 10.3390/nu15224851

**Published:** 2023-11-20

**Authors:** Joseph R. Sharkey, Andra Smith

**Affiliations:** 1School of Public Health, Texas A&M University, College Station, TX 77843, USA; 2Sequim Food Bank, Sequim, WA 98382, USA

**Keywords:** parent–child intervention, experiential intervention, child cooking and food preparation, vegetable preference, rural populations, cooking curriculum, vegetable exposure

## Abstract

Although children from limited-resource families in rural areas are at great risk for nutrition-related chronic diseases, few hands-on programs have been implemented that simultaneously engage both parents and children and include local produce in a single program. This study reports on the development, implementation, and evaluation of Cooking with the Seasons for Health (CwS4H). Parent–child pairs participated in six sessions (two weekly sessions during each of three growing seasons), which included food tasting, a spotlight vegetable, interactive mini nutrition lesson, a child-focused cooking lesson, hands-on meal preparation, distribution of materials as family guides, and a take-home bag of fresh produce. Pre- and postprogram survey data were collected from 23 parents and 22 children. Children reported improvements in nutrition knowledge, vegetable preference, and self-efficacy in food preparation and cooking. Parents reported gains in nutrition knowledge, nutritional behaviors, vegetable preference, attitude toward food preparation/cooking, involvement of the child in food preparation/cooking, confidence in preparing vegetables, and the child’s vegetable intake. Parents commented on the value children placed on food preparation and produce selection and how the program enhanced the parent–child relationship. By focusing CwS4H on a variety of fresh vegetables, this intervention helped to impact children’s vegetable intake behaviors by engaging children in preparing and choosing the food they eat.

## 1. Introduction

The burden of chronic disease, especially overweight and obesity among children, disproportionately affects marginalized populations, such as children from limited-resource families who reside in rural areas. Childhood obesity is associated with a multitude of health problems such as cardiovascular disease, type 2 diabetes, and other metabolic diseases [1,2]. Further, diet and nutrition behaviors established in childhood have been associated with the risk of chronic disease later in life [3,4]. Interventions that focus on elementary-school-age children can impact fruit and vegetable (FV) behaviors before they are established [5]. There is substantial evidence that a single dietary change, such as increasing FV consumption, is important and has been associated with a decreased incidence of and mortality from a variety of chronic diseases: diabetes, cardiovascular diseases, stroke, hypertension, obesity, and certain types of cancer [6,7,8,9,10]. Specifically, vegetable intake is correlated with the Healthy Eating Index score, which is a measure of dietary quality and includes dietary recommendations, and children’s health and weight status [11,12]. This is especially critical for low-income children and youth in limited-resource rural areas, who are at increased risk for developing chronic disease and having to manage it for a lifetime [13].

Children’s preferences and eating habits are complex and involve individual, family, and environmental influences. There are interrelated factors that influence vegetable intake among children: vegetable exposure, preference for vegetables, acceptability and liking of vegetables, knowledge, attitudes, self-efficacy, parental involvement, and a supportive environment [3,12,14,15,16,17,18]. One strategy for expanding exposure to new and different vegetables, preference, acceptability, and consumption of vegetables is to involve children in the process of meal preparation [11,19,20,21,22,23,24,25], which provides children with greater confidence in food skills, cooking practices, cooking attitudes, and diet quality, which supports positive cooking-related behaviors and higher diet quality later in life [17]. Children have the developmental capacity to learn cooking skills from younger ages [26]. Cooking workshop programs that involve children in cooking can increase their willingness to taste novel foods and direct food choices toward foods containing vegetables [18]. Overall, cooking programs have a positive impact on the beliefs, knowledge, skills, preferences, attitudes, and behaviors around nutrition and cooking in children, with reported high enjoyment in participating in cooking programs [23,27].

Experiential learning is described as a process that is more engaging to children compared to more traditional learning approaches [28]. Hands-on experience with food is an engaging and effective strategy to teach healthful eating behaviors [29]. In a systematic literature review, Varman et al. found that experiential approaches, such as cooking, preparing food, and taste testing, increased children’s willingness to taste unfamiliar foods [28]. Studies have found that hands-on cooking skills can provide motivating experiences that influence children’s eating behavior [23,25]. One explanation for the value of cooking skills and food preparation for children is the value they place on their creation, often referred to as the “I cooked it myself” effect or “IKEA effect”, where there is an increased liking of self-prepared foods leading to higher consumption [19,25,30,31]. 

A systematic literature review by Charlton et al. identified characteristics of successful primary-school-based experiential interventions, which included cognitive-based outcomes (nutrition-related knowledge, preferences or attitudes, and self-efficacy), a focus on increasing vegetables, sending home fresh food along with recipes, repeated taste testing, frequent exposure that included multiple experiences, and the involvement of parents [3]. There have been several different approaches to cooking intervention. One approach is to target parent–child dyads, such as in Fun with Food [17] and Cooking Matters for Families [21], which increased children’s perceived cooking competence and parents’ comfort with allowing children to participate in kitchen activities. Cooking Matters for Families focused on addressing procuring vegetables, using various vegetable preparation methods, and incorporating vegetables into meals and dishes. The results of this program included increased parental cooking confidence, healthy food preparation, child self-efficacy, vegetable variety for the parent and for the child, and home vegetable availability [21]. It was shown that parents’ involvement of their children in cooking activities impacted children’s liking of vegetables, which impacted vegetable intake [24]. A second approach focused on children in school-based programs. Cooking with Kids exposed children to foods from different cultures and indicated increases in vegetable preference, cooking attitudes, and self-efficacy [29,32,33]. Common Threads [34] provided experiential cooking and nutrition education, with a focus on cooking skills through a chef-led afterschool program. This program improved cooking self-efficacy, exposure to new foods, and liking and consumption of vegetables. Texas, Grow! Eat! Go! [35] used hands-on cooking involvement and cognitive factors such as attitudes and self-efficacy as a mechanism for increasing vegetable preference, vegetable exposure, and a positive relationship between family cooking and vegetable intake. A third approach supplemented skills and education with the use of local agriculture, such as in Farm Fresh Foods for Healthy Kids (F3HK), which used local produce in the form of community supported agriculture (CSA) with skill-based and seasonally tailored healthy-eating classes [9]; Flint Kids Cook, which is set in farmers’ markets [36]; or Brighter Bites, which distributes fresh produce through a school-based food co-op along with a nutrition curriculum [37]. In addition, this paper describes the experiences of parents and children in CwS4H.

## 2. Materials and Methods

### 2.1. Program Overview, Study Design, and Context

Program overview: CwS4H was an innovative experiential nutrition education program for parent/caregiver (referred to as parents in this article)–child dyads that linked participating families to local farms through a community supported agriculture (CSA) approach in the form of Good Food Bags (GFBs), utilizing seasonal produce from local farms during three distinct growing seasons with different produce, children’s hands-on preparation and cooking of the produce in the GFB, recipe cards, take-home materials, and knowledge sessions for adults and children. Two community champions were hired to recruit parent–child pairs for two 2.5 h sessions during each of three distinct growing seasons (a total of six sessions) in 2018. Pre- and postprogram surveys were separately administered to participating parents and children. Children took the lead in food preparation and in personally assembling their GFB at the end of each session. Study design: A mixed-methods approach was used, which included a single-group quasi-experimental design (pre- and postprogram surveys) followed by focus groups. Context: The CwS4H program was developed by an academic–community team as part of long-standing community collaborations, including with local food banks, area farmers, and other community members. The collaborations included academic researchers in public health with nutrition expertise, the executive director of the local food bank, and community members with expertise in food preparation and nutrition education. CwS4H was implemented in two rural communities (Sequim and Port Angeles) on the North Olympic Peninsula of Washington. In Sequim, programs were held on Sunday afternoons and Monday evenings (two local churches) and in Port Angeles on Saturday mornings and afternoons (Lincoln Skills Center). All locations provided a commercial-grade kitchen and space for hands-on work areas, nutrition education, and group sessions. 

### 2.2. Participants

Participants included area farmers and parent–child pairs in two rural communities. Farmers: Area farmers and growers were contacted by the team to determine farm size, location, produce plans, and willingness to work with the program. The executive director of the Sequim Food Bank (author AS) has an established relationship with many of the local farms and served as program lead for farmer recruitment, produce acquisition for CwS4H, and produce transportation to program sites. In conversations with the farmers, AS explained the program, obtained a list of produce currently growing and estimated time of harvest, and created a mutually agreed-upon price list. Six farmers recruited were from small- to medium-size family farms that were established in four rural areas. Parent–child pairs: A parent and one child (3rd–5th grade) were invited through flyers, handouts, Facebook, and direct contact with service providers to participate in a six-session program (June–October) to learn about ways for the parent and child to be creative in the kitchen and prepare tasty, affordable, and healthy meals using fresh local produce. Participants were asked to commit to participate in all six sessions, of which two sessions were to take place in each of June, August, and October, and complete a survey before and after the program. Parents provided written consent for their participation and permission to approach their child; the child provided assent to participate. As an incentive for participation, participating parent–child pairs received a free “Good Food Bag” of fresh produce at the end of each of the sessions to take home. At the end of the program, the participants received a kitchen tool kit. Parent–child pairs were recruited for specific sessions (e.g., Saturday mornings or afternoons in Port Angeles, Sunday afternoons or Monday evenings in Sequim). 

### 2.3. Theoretical Foundation

CwS4H drew on several theories, such as social cognitive theory (SCT) [38,39], experiential learning theory (ELT) [28,40], and the family ecological model [41], to provide skill-based and seasonally tailored education and hands-on activities to support acceptance, utilization, and consumption of fresh vegetables in children and parents/caregivers based on ELT. Guided by SCT, CwS4H addressed attitudes and beliefs about the value of consuming vegetables (outcome expectation), improved skills and self-efficacy with respect to selecting and preparing local produce (self-efficacy), reduced barriers to the acceptance of local vegetables and to developing strategies for increasing consumption of vegetables (barriers), provided opportunities for participants to observe peers demonstrating newly acquired skills and share experiences via group discussion (observational learning/modeling), and enhanced the value of cooking skills and food preparation for children through their perceived value of their own food creation [19,25,31]. By engaging both the parent and the child in CwS4H, the FEM accounted for parent–child communication and the family environment. 

### 2.4. Program Structure

CwS4H was a six-session nutrition program, with two weekly group sessions during each of three growing seasons. One week prior to each session, participating farmers were contacted by AS for available produce, orders were placed, and pick-up was scheduled for harvested produce. On the Friday before each of the weekly sessions, the lead author picked up and transported all program produce to refrigerated storage. All session produce was delivered to the session sites one to two days prior to each session. Session components included in-person group sessions and at-home activities, which focused on procuring fresh vegetables, using various vegetable preparation methods, and incorporating vegetables into meals and dishes [21]. The program’s overall theme was enhancing the experience of parent–child pairs cooking together while promoting a sense of connectedness, providing a positive social experience, and improving children’s skill building. Table 1 presents the six session themes.

### 2.5. Program Sessions

Every CwS4H group session included food tasting, a spotlight vegetable, an interactive mini nutrition lesson, a child-focused cooking lesson, hands-on meal preparation, eating together and recap, distribution of materials for a family guide, and the assembly of a Good Food Bag (see Table 2 for an example of week 1). Tasting recipe lesson: Each session included 2–3 welcome tastings (drink, side, snack) for each parent–child pair to sample. Two handouts were provided to each family for each tasting: (1) an information sheet of ingredients, a recipe, mix-it-up directions, tips, and health benefits, and (2) the “Everyone in the Kitchen” sheet, which identifies the specific tasks required to complete the tasting recipe along with who can perform the tasks—children with parental supervision or children alone. Figure 1 and Figure 2 show examples from session 1. Spotlight on vegetables: During the session welcome, spotlight vegetables were introduced and handouts provided that describe what the vegetable is, its nutrition benefits, its health benefits, how to use it, how to prepare it, and fun facts about the vegetable. Figure 3 and Figure 4 show two examples of spotlight vegetables (kale and kohlrabi). Interactive mini nutrition lesson: Table 3 outlines the nutrition minilessons and key nutrition messages, which the team guided participants through; the team also provided them with colorful handouts. For example, in session 1, the team introduced MyPlate, the importance of food safety, tips for washing and storing vegetables, and cooking with more vegetables. Activities included two “challenges”: vegetable identification and a handwashing competition. Child-focused cooking lesson: All steps for the session’s main recipe and skills necessary were demonstrated. Information was tailored to individual parents and children. The recipe was linked to nutrition education and health. Materials were presented to children in an engaging format (brightly colored logo, pictures, and icons). Hands-on meal preparation: Parents and children learned to procure, prepare, and serve vegetables at mealtime and for snacks. Individual workstations for the children with child-safe kitchen and cooking utensils needed to complete the recipe were set up. Children were responsible for the meal preparation, with their parent behind them to advise. Using the recipe, children selected their items from a common table. Parents were encouraged to provide positive reinforcement for child-engaged activities. Figure 5 and Figure 6 show the information sheet and “Everyone in the Kitchen” sheet for the main recipe in session 1 for hands-on meal preparation. Eating together: Eating together included a guided group discussion as everyone enjoyed the food they had prepared. Children and parents discussed their in-session experiences and how they could practice the skills before the next session. Each family received a binder for program materials. Starting in session 2, participants were asked “What they did since the last session?” They were encouraged to describe food dishes prepared and any recipe modifications. Good Food Bag: CwS4H tapped into three different growing seasons and provided available produce. Each child handpicked the planned amount of each produce item from produce boxes and placed the produce in their personal Good Food Bag (GFB). Instead of preassorting the produce, the team determined the importance of each child hand-selecting the produce for their family’s Good Food Bag. In preparation for produce distribution, AS wheeled out a cart that had one of everything to be placed in the GFB and then held up and talked about the produce item in terms of which farm produced the item and shared an anecdote from the farm. Table 4 lists the Good Food Bag produce for sessions 2, 4, and 6.

### 2.6. Evaluation

The trained program staff separately collected all data from each parent and child. Pre-program surveys were interviewer-administered prior to the start of the program and post-program surveys at the conclusion of the program. Focus groups with children and focus groups with parents were conducted after the completion of the program, and individual interviews were conducted with participating farmers. A complete list of outcome variables and food preparation and cooking questions for children and parents are shown in Table 5. Visual analog scale cards were used to assist with responses to survey questions. Child surveys: Survey data collected both pre- and post-program include nutrition knowledge, vegetable preference/liking, vegetables never tried, self-efficacy in food preparation and cooking, vegetable intake yesterday, and parent role modeling. Data on four topics were collected in only the preprogram survey: (1) willingness to try new foods, (2) attitude toward cooking, (3) frequency of assisting a parent, and (4) confidence in preparing or eating vegetables. After completion of the program, data were collected on vegetables tried in the past month and food activities performed in the past week. Parent surveys: Preprogram and post-program surveys collected data on participant characteristics, nutrition assistance program participation, nutrition knowledge, nutrition behaviors, vegetable preference/liking, attitude toward food preparation activities, positive attitude toward cooking, time as a cooking barrier, family food practices, confidence in preparing vegetables, vegetable intake, and family support for vegetables. Data collected post-program included confidence in performing food activities, vegetable availability in the household, child participation in food activities, and household food security.

Unstructured observations were completed by team members during each session, followed by post-session debriefs/feedback with team members. Although not formally part of the evaluation, parents and children provided feedback prior to and after each session, and parents shared with the team their weekly Facebook posts of home food preparation and meal activities. Parent focus groups were conducted by the first author approximately one month after the completion of CwS4H in a meeting room at the locations of the CwS4H sessions. The guiding questions focused on the CwS4H lessons, the child’s health, and the value and community. Child focus groups utilized the draw, write, and tell (DWT) method [42,43,44]. The guiding questions of DWT were as follows: (1) “Draw -your-cooking”, focusing on a particular weekday and weekend day, which includes cooking/food preparation activities for meals and snacks; (2) write on the pictures what the characters or pictures are thinking or saying; (3) tell me about your drawing; (4) describe the drawing; (5) explain what the drawing means; (6) explain why you decided to draw those images; (7) tell a story; (8) provide a title for the drawing; (9) describe how this picture would be different if drawn before you participated in CwS4H. Finally, children were asked to talk about their experiences with the CwS4H program. Interviews with farmers were conducted at each farm location. Three areas were explored with the farmers: (1) background information on their farm, (2) experiences with the CwS4H program, and (3) suggestions for future participation.

### 2.7. Data Analysis

*Cooking with the Seasons for Health* was a pilot experiential nutrition and cooking program for parent–child pairs. Survey data for parents and children were separately examined with descriptive statistics and within-person change from preprogram to post-program using Stata 12 at a significance level of *p* < 0.05 [45]. Frequencies were calculated for participant characteristics. Analysis of change between pre- and post-test were performed using a paired samples t-test and Bonferroni correction for multiple comparisons to estimate the significance between the means of the two samples (pre- and post-test) from the same participants. With the correction for multiple comparisons, the significance level pair *t*-tests among children (5 *t*-tests) was *p* < 0.01, and among parents (8 *t*-tests) was *p* < 0.006. Written field observations for each session were reviewed. Focus group and interview data were audio-recorded, transcribed verbatim, and interpreted using the *Sort and Sift*, *Think and Shift* qualitative data analysis approach [46]. This approach involves an “iterative process where analysts dive into data to understand its content, dimensions, and properties, and then step back to assess what they have learned and determine next steps”. Sorting and sifting involved reading the manuscripts, reviewing, and recording observations, and thinking and shifting required reflection, re-strategizing, and reorienting [46].

## 3. Results

### 3.1. Sample Characteristics

Thirty parent–child pairs completed the preprogram survey and started the program: 53.3% (*n* = 16) in Port Angeles and 46.7% (*n* = 14) in Sequim. Among the 30 participating children, 56.7% (*n* = 17) were boys and 43.3% (*n* = 13) were girls. A total of 4 children were accompanied by their fathers (3 were girls) and 26 children by their mothers (16 were boys). Demographic characteristics reported by parents are shown in Table 6 and nutrition assistance program participation in Table 7. All 30 pairs participated in the first two growing seasons (sessions 1–4). Sessions 5 and 6 in growing season 3 (fall) and the post-program survey were completed by 23 parents and 22 children: 92.9% (*n* = 13) of Sequim and 62.5% (*n* = 10) of Port Angeles participant pairs. All the fathers (100%) and 73.1% (*n* = 19) of mothers completed both preprogram and post-program surveys, along with 76.5% (*n* = 13) of boys and 69.2% (*n* = 9) of girls. There was no statistically significant difference between completers and non-completers in parent or child gender, parent marital status, parent or child age, parent completed education, or household composition. We did observe that in the case of Port Angeles, where all sessions were conducted on Saturdays, children’s or family activities may have affected completion, and two families transferred out of the area for work during the program. Data reported in the following sections include the 22 children and 23 parents who completed both preprogram and post-program surveys.

### 3.2. Children Surveys

Pre- and post-program survey results are shown in Table 8. A total of 22 children completed both pre- and post-program surveys. As shown in the table, improvements in nutrition knowledge (*p* = 0.016) were observed in mean total from preprogram (3.77 ± 1.44) to post-program (4.36 ± 1.25). Although vegetable preferences (like) increased from pre- to post-program, the difference was not statistically significant (*p* = 0.248). Six vegetables were added at post-program, and the total of liked vegetables ranged from 1 to 6 (3.54 ± 1.68). For the two fall vegetables, 40.9% (*n* = 9) liked kohlrabi and 72.7% (*n* = 16) liked pumpkin. Considering the Bonferroni correction at *p* < 0.01, there was a borderline significant (*p* = 0.015) decrease in the mean total of vegetables that children had never tried or were not sure if they tried between preprogram (1.82 ± 1.84) and post-program (0.95 ± 1.33). Self-efficacy in food preparation and cooking increased significantly (*p* = 0.002) from preprogram (8.00 ± 2.41) to post-program (10.14 ± 2.42). Parental modeling remained high at both survey times (*p* = 0.648), with 68.2% (*n* = 15) reporting all four activities at preprogram and 72.7% (*n* = 16) at post-program. Four topics were asked of children only in the preprogram survey: (1) among the nine situations regarding willingness to try new foods, the mean reported was 4.5 (±1.63); (2) children reported a total mean of 2.41 (±1.33) of the four attitudes toward cooking activities; (3) weekly assisting of the parent in helping prepare dinner was 22.7% (*n* = 5) and asking to go with the parent to the grocery store was 45.4% (*n* = 10); and (4) the mean total of six activities (2.04 ± 1.91) described a high level of confidence (very sure) in preparing and eating vegetables. Of interest, among the four attitude-toward-cooking activities, 81.8% (*n* = 18) felt really good making food with the family, 72.7% (*n* = 16), felt really good about the food they helped cook, 45.4% (*n* = 10) felt really good making snacks with vegetables, and 40.9% (*n* = 9) felt really good about the taste of vegetables. Two new topics were included in the post-program survey: a total of 23 vegetables tried in the last month (13.27 ± 3.76) and 13 food activities performed (9.18 ± 2.30).

### 3.3. Parent Survey

The results from the parents’ pre- and post-program surveys are shown in Table 9. A total of 23 parents completed both surveys. Statistical significance for the parent survey, with adjustment for multiple comparisons, was set at *p* < 0.006. Parents reported a significant gain in nutrition knowledge at the completion of the program (*p* = 0.002). Although nutrition behaviors increased from preprogram (2.30 ± 1.22) to post-program (2.61 ± 1.47), the results were not statistically significant. In the preprogram survey, 21.7% (*n* = 6) reported that they never eat food past its “use by” date, which increased to 34.8% (*n* = 8). The increase in always checking that food is piping hot when reheating increased from 47.8% (*n* = 11) to 60.9% (*n* = 14). There was a small increase in the number of vegetables that parents liked from preprogram to post-program. During the preprogram survey, 82.6% (*n* = 19) of parents liked at least 12 of the 16 vegetables, which significantly increased (*p* = 0.026) to 95.6% (*n* = 22). Five additional vegetables were included in the postsurvey, iceberg lettuce, kohlrabi, garlic, celery, and pumpkin, with parents liking 3.83 ± 0.98 of the five vegetables. The least liked was kohlrabi. Parents were also asked in the post-program survey which of 21 vegetables they tried in the past month. More than half of the parents tried at least 18 different vegetables in the past month (56.5%, *n* = 13). At post-program, parents tried 13.27 (±3.76) of 21 specific vegetables, with 56.5% (*n* = 13) having tried at least 18 different vegetables in the past month. Attitude toward 13 food preparation activities: parents exhibited significantly (*p* = 0.001) increased positive attitudes from the preprogram survey (8.69 ± 3.05) to the post-program survey (*p* = 10.74 ± 2.47). For example, a greater percentage of parents did not feel limited by nutrition knowledge at post-program (65.2%, *n* = 15), compared with 39.1% (*n* = 9) of parents at preprogram. Positive attitude toward cooking: although not statistically significant (*p* = 0.287), parents increased the number of moderately or strongly positive attitudes. For example, parents who found cooking to be a fulfilling activity increased from 56.5% (*n* = 13) to 73.9% (*n* = 17), and parents who found cooking to be not a waste of effort increased from 69.6% (*n* = 16) to 78.3% (*n* = 18). Time as a cooking barrier: parents experienced fewer examples of time as a barrier to cooking at post-program (*p* = 0.417). Specifically, fewer parents reported needing more time to plan for meals (47.8% vs. 56.5%) and finding enough time to prepare preferred foods (39.1% vs. 56.5%) at post-program compared with preprogram. Family food practices: involvement of the child in food-related activities at least once a week increased (*p* = 0.036) from preprogram (3.30 ± 1.02) to post-program (3.83 ± 0.89) but lacked statistical significance after adjustment for multiple comparisons. Parents increased talking with their child about eating healthy foods at least once a week from 78.3% (*n* = 18) to 95.6% (*n* = 22), and their child helping prepare dinner at least once a week increased from 43.5% (*n* = 10) to 56.5% (*n* = 13). At post-program, 52.2% (*n* = 12) of parents included their child in meal planning at least once a week. Confidence in preparing vegetables: although not statistically significant (*p* = 0.447), parents reported increased confidence in preparing specific vegetables. Child’s vegetable intake: on a normal day, the parent’s child consumed vegetables one to two times; 91% (*n* = 21) at preprogram and 95.7% (*n* = 22) at post-program. Family support for vegetables: all parents at post-program identified their spouse and other family members in providing support for making it easier for the child to eat vegetables. Regarding lack of support, the child’s siblings or no one in the household did not support making it easier for their child to eat vegetables. All parents identified the health benefits of having vegetables in the household. Confidence in performing food activities: parents reported moderate or extreme confidence in performing 12 different food-related activities (9.48 ± 3.15), with 73.9% (*n* = 17) confidence with at least 9 different activities and 56.5% (*n* = 13) with 11–12 activities. Vegetable availability: the almost-every-day availability of vegetables in the household at post-program included (1) vegetables in my home (95.6%, 22), (2) vegetables are part of my child’s meal (82.6%, 19), (3) vegetables are given to my child as a snack (65.2%, 15), (4) there are cut-up vegetables in the fridge for my child to eat (30.4%, 7), and (5) I try to get my child to eat more vegetables (82.6%, 19). More than 50% of parents (52.1%, *n* = 12) identified that at least four of the five items were available almost every day. Child activities: after completing the program, parents reported on 11 specific activities that their participating child does on his/her own (9.1 ± 1.9), with a range of 5–11. The participating child participated in the following activities on his/her own in the past week: make snack with vegetables (78.3%, *n* = 18), help make a family meal (78.3%, *n* = t 18), cut up food (82.6%, *n* = 19), measure ingredients (65.2%, *n* = 15), use a can opener (39.1%, *n* = 9), try new foods (47.8%, *n* = 11), and clean or wash FV before using (73.9%, *n* = 17). Food security: At post-program, 56.5% (*n* = 13) of parents reported at least one food security item in the past month. Specifically, 26.1% (*n* = 6) worried that they would run out of food before they received money to buy more, 17.4% (*n* = 4) of households ran out of the foods needed to make a complete meal, and 17.4% had to choose between paying bills or buying food.

### 3.4. Children’s Focus Groups

Seventeen children participated in three focus groups that took place on 17 November 2018 (Port Angeles 1) and 4 December 2018 (Port Angeles 2 and Sequim). The three sessions were conducted using the draw, write, and tell (DWT) method. Key themes included the following: (1) overall positive experience, (2) bonding with parent, and (3) helping prepare meals. The overall experience was best described by one of children: “Also, my mom and dad say What’s my number one rule? Have fun”. Another child exclaimed that “I never wanted it to end and I hope it comes again next year”. The drawings produced by the children described food-related activities on weekdays and weekend days. One child reported: “My drawing is about the weekdays where I sit at the table and drew about my mom’s famous potato soup. I said, Mom this is yummy”.

Children described their feelings toward cooking with a parent as “bonding with our parents”. Another child talked about bonding: “My parents trust me to be in the kitchen more, and we started to bond more and have family cooks and stuff. I’ll cook with my dad a lot”. Additional children discussed what the program provided them. One child stated that “I am spending more time in the kitchen and what I am doing is helping my mom bake by doing just about everything, making snacks, and I am cutting just about everything I can cut”. According to another child, “I’ll say something, so my dad taught me how to cook and that was our special time together because he worked a lot so he didn’t have a lot of other time. So, I made dinner with him and that was my favorite memory with him, and now I cook all those same foods”. One child mentioned that as a result of the program, “I have been able to use knives more, so I have been able to help my dad cut things”.

### 3.5. Parents Focus Groups

Four topic areas were discussed by the 14 parents that participated in three focus groups, which were held at the same time as the children’s focus groups. The topic areas were reasons for participation in CwS4H, benefit of cooking with the child for the child’s confidence, relationship with child, and the Good Food Bag. One parent stated that the program “forces me to slow down, let her (my child) come in and let her do it. Now I can just send her into the kitchen to do it. So, it’s a blessing to me!” Another parent talked about her son’s experience: “Here they’re (the children) willing to try. My son has been having a hard time to try new stuff, but somehow, he tries almost everything here! I’m like, “Sweet!” And that’s just opened up to, you know, buying new vegetables and recipes. It really helps”. Parents reported that because of the program, “it gave me the opportunity to give him (my child) more liberty in the kitchen because I am more comfortable with what he is doing”.

Parents recognized that there were many benefits of the CwS4H for their children: a key theme was *child is now in charge*: “The confidence really soared, like you said. Because I was trying to help, he was like, “Mom, scoot back, I got this!”” Several parents talked about their child taking charge; for example, “like I’m just the assistant and he’ll do most of it. I just have to sit back and observe”. Parents stated that “he (child) is making menu choices now which is nice. Another parent confirmed this by describing the child saying “Mom can we have this vegetable with dinner?” or “ Mom, I like broccoli now!”. The parent replied, “and I’m like, “Really? Alright”. Cooked broccoli only, but the cooking is done by him and since he has cooked it and tried it, he’s a lot more open to the cooked foods and cooked vegetables”. A second theme was *I (child) can do this*. One parent stated, “it was seeing her here and having you guys kind of support and the other kids are “I guess the other kids are cutting stuff. I can let Jordan (my child) cut stuff”. It’s given me more confidence in Jordan, and in her having more confidence also. She’s got good self-esteem from it.” One parent stated that “now she’s just opened her up to choices she never would have done before. The tasting it here thing I feel like lessened the scariness of it”. Another parent exclaimed that “my daughter is so more open to eating everything now”.

A third and major theme was *relationship with the child*. One parent confided that “I definitely think it has taken a lot of tension out of the kitchen because now she comes in eager to help instead of me dreading it, I am eager to accept her help. She has even had ideas for recipes and is now helping me make the list for grocery shopping, plus she is really excited about it, so she tries to teach her brother”. Parents described how the CwS4H program affected their relationship with their child. One parent stated that “I wanted to spend more time with my son and try and encourage him to like a greater variety of foods”. Another parent described that “what I got out of was like the bonding”. An additional parent explained that “I chose to participate in this program because I wanted to spend more time with my daughter”. Parents explained that the CwS4H classes provided an opportunity for the parent and child to bond. As one parent explained, “here she was mastering the big knife and loving that and then it was also something that we could do together to bond together, too. Sometimes we have a lot of difficulty connecting in the right areas, and this was something that we can do together”. Another parent shared that “we don’t get really get that much one-on-one time because he goes to his dad’s house every other weekend, so I really want to spend time with him every other weekend and its with everyone else. It was important to get some time alone”. A parent commented that “we absolutely enjoyed every class together. He has come home so excited to tell his dad all about it, about making new things. It’s been really fun for us. It’s mainly the together time to actually do something together. There’s not a lot of opportunities to do a parent-child class”. An additional parent summed it up with the following: “It (the program) has become this thing that we do together that has been really nice”.

A fourth theme centered on the *Good Food Bag*. As a result of the Good Food Bags (GFBs), parents described many benefits of the produce that the children selected for their GFB. Parents explained that “a lot of the vegetables the kids didn’t even know existed, so to be able to recognize what they have seen in class to what they have in their good food bag at home reinforces the identity of those vegetables”. Another parent mentioned that “being able to bring the good food bag home, have the recipes on hand, and go from there, that was huge! It was a really big help, and then there was farms that I didn’t even know about”. Several parents commented on the children taking ownership of the produce they selected for their own GFB. One parent stated that “Well I also think it helps because they have chosen it. They get to put it in the bag. They get to bring it home and share it. There is an ownership there that isn’t there when we go to the grocery store”. On a funny note, parents commented on how “one bag is way too heavy” and the child dragged it to the door. Another parent explained that “Mine (my child) likes to make things for dad from the good food bags. He tells his dad what was made in class today and then we will let him make it for dinner or something during the week. It has involved some pride for him and confidence building in the kitchen”.

### 3.6. Farmer Interviews

Individual interviews were conducted in December 2018 and January 2019 with farmers from five farms that participated in CwS4H. Joy Farm and The Farm were two small, family farms in Sequim. Joy Farm grows produce on about 2 acres and started in 2016. The Farm grows produce on ½ acre and also raises chickens. Johnston Farm is located between Sequim and Port Angeles, farms 4.5 acres, and has farmed since 2000. Wild Edge, which is located West of Port Angeles, consists of 89 acres that are largely wildlife habitat, hence the name Wild Edge, and grows produce on less than 1 acre. Reaume Organic Farm uses 3 acres for production in the Beaver/Forks area (about one hour west of Port Angeles). Farmers talked about “the time it took for harvesting, the washing, and all for the project”. However, one stated, “I think one thing was to help people to know what is available in the area”. The Joy Farm husband and wife team completed undergraduate degrees in agroecology, a program “that applies ecological principles to agriculture”. They further described insights from participating in the program:

“It was meaningful and eye-opening seeing the need that’s there in that community, and we also have always felt, you know in our education, we not only learned about agriculture, but we were also learning about social justice and social issues surrounding agriculture, access to food is one of them. So, that’s something that we’ve always been aware of, and it’s difficult finding the balance between farming our livelihood, and making ends meet on that end, and also trying to make sure that our food is getting to people of all means, and that is a tricky balance, because making a living farming isn’t easy.”

“One thing that this program did, is it allowed us to, I guess it encouraged us to grow a diversity of things. There were several things we wanted to try that we didn’t necessarily have a big market for, and we weren’t expecting that this program would buy our full harvest, but we thought at least we’ll sell some of it. So, it gave us a little bit of rationale for saying, “Okay, let’s plant a couple rows of cucumbers and see how they do”. And the cucumbers did awesome, we were, that’s something that we might not have devoted space to if we had zero market for it.”

“As a farm that’s getting started it allowed us to grow new and different things, knowing that we would have at least some market for it. Then having the product it’s easier to find customers, rather than approaching people and saying, “hey, can I grow this for you?” But yeah, it definitely helped to open a lot of doors for us.”

A more rural farmer explained the challenges of weather, which affect when they can begin planting. They talked said, “elks got their own good food bag. That’s part of just life. I think about that, it’s cooking with the seasons. Well, that’s a season, the elk are down because they’re looking for food. They came across some yummy carrots, so yeah”. Another farmer stated that “I think that it is a great program that you are helping farmers make money and supporting them instead of buying it from Charlie’s. Which would be a lot easier for you and cheaper to buy from a giant farm. I think it is really great that you support the local farms because it helps the community get better”.

## 4. Discussion

In this quasi-experiment, we found that *Cooking with Seasons for Health* (CwS4H) strengthened parent–child dyads’ knowledge of, attitudes toward, preferences for, and preparation skills for the variety of vegetables grown during three different growing seasons. Unique to this program was the integration of the following components in six sessions during three distinct growing seasons: (1) a focus on increasing consumption of a variety of vegetables critical in the prevention chronic diseases, such as type 2 diabetes, cardiovascular disease, and some cancers [14,47]; (2) using Good Food Bags, an adaptation of a food co-op/CSA approach, which is an appropriate mechanism for offering a variety of local, high-quality seasonal produce and an important component in a comprehensive behavior change program [48,49]; (3) linking vegetable distribution with interactive nutrition education and practical food preparation skills to facilitate improvement in education, skills, competency, and behaviors [37,48,50,51]; (4) targeting environmental factors (availability and accessibility in the home, family and peer influences), behavioral factors (parents’ knowledge of intake recommendations and skills, child involvement in preparation of meals and snacks, pre-preparation of vegetables for children), and personal factors (food preferences, vegetable preferences, and preferred preparation styles) to increase children’s vegetable behaviors [17,29,52,53]; (5) applying aspects of learning theory, including social learning, liking and intake of vegetables, and parents supporting children’s skill building [54]; and (6) education and skill building taking into account influencing factors and strategies needed to making vegetable consumption behavior habitual [55].

Further unique aspects of this program were the engagement of parent–child dyads in two rural communities and the exposure of all participants to the vegetables locally grown during three growing seasons. Four groups of parent–child dyads (*n* = 30 pairs) participated in two rural communities. In six sessions (two sessions for each of three growing seasons), CwS4H introduced the same participants to different produce available during three distinct growing seasons. Hands-on nutrition education and food preparation provided parents and children the opportunity to interact with foods and engage in the cooking process. While 30 parent–child pairs completed the baseline survey, 23 parents and 22 children completed the three-season program and the pre- and post-program surveys. Using pre- and post-tests, we found that participating children reported significant improvements in nutrition knowledge of and self-efficacy in food preparation and cooking, and a shift toward increased vegetable preferences that was not statistically significant. The results among parents included the following: a highly significant gain in knowledge, increased nutrition behaviors and vegetable preferences, significant positive attitudes toward food preparation activities and cooking, a significant increase in family food practices, and fewer examples of time as a limitation for cooking behavior. Data collected from parents at the completion of the program indicated a broad availability of vegetables in the home and their child participating in a large number of food-related activities during the previous week. Children were willing to try vegetables when preparing and eating with the parent and other children, especially when they were unfamiliar with the vegetable. The results extend previous work that demonstrated that experiential approaches, such as hands-on cooking skills, food preparation, and taste testing, increased children’s willingness to taste unfamiliar foods and influenced children’s eating behavior [23,25,28]. One explanation for the value of cooking skills and food preparation for children is the value placed on the food’s creation, often referred to as the “I cooked it myself” effect or “IKEA effect”, where there is an increased liking of self-prepared foods leading to higher consumption as the child takes ownership [19,25,30,31]. Another explanation recognizes the importance of parents’ encouragement in involving children in the preparation of healthy meals, as this improves the liking of vegetables and, thereby, increases the children’s vegetable intake [24].

Focusing interventions like CwS4H on a variety of fresh vegetables impacts children’s vegetable intake behaviors before they are established, especially those that engage children in preparing and choosing the food they eat [5,19,20,21]. Key to this program was experiential learning, which used multiple strategies involving both parents and children and increased children’s willingness to taste unfamiliar foods [28]. As previously mentioned, CwS4H was implemented using an integration of multiple characteristics of successful programs with children: health-related and cognitive-based outcomes (e.g., knowledge, preference, self-efficacy); sending home fresh, locally grown vegetables; hands-on cooking lessons; repeated taste testing; parent involvement; frequent exposure to a variety of seasonal vegetables; and food preparation skills [3,23].

Our results in two rural communities with limited-resource families confirm the results of Fun with Food, a four-week parent–child cooking intervention, using a quasi-experiment pretest post-test design [17], which accomplished the following: (1) enabled parents to become more comfortable allowing their children to take part in kitchen activities and reduced the fear of introducing cooking skills; (2) increased children’s perceived cooking competence and interest in cooking; (3) changed parental perceptions about including children in cooking; and (4) added the benefit of the parent and child spending time together. The overall favorable view of the CwS4H program was likely due to children bonding with their parent and building parent trust, the heavy and highly varied GFB, and the introduction of so many different vegetables that were new to them. For parents, the program enhanced confidence for the child and with the child, a greater appreciation for local farms, an improved relationship with the child, and enjoyment for both—that they “do this together”. CwS4H focused on parent–child pairs in a way that valued their collaboration and mutual mentorship.

This program is unique in a number of ways, namely, the parent and child spending quality time together with the parent serving as mentor; the inclusion of three growing seasons; the use of Good Food Bags; and being child-directed (cooking, tasting, Good Food Bag composition). Further, there appeared to be a confirmation of the “IKEA effect” in children, showing that when children cook together with their parents, children’s liking of vegetables and vegetable intake increase [24]. Parents observed that children took ownership. For example, children picked the food for the Good Food Bags, and a parent said, “you know that was their pumpkin, or their squash, or their kale. They took ownership of it to feed the family when they got home. I thought that was a great, something added to what we were doing”. The parents noticed that and commented that these were the kinds of food they would cook when they got home during the week, because their child would say, “I picked that squash, I picked that broccoli”.

The program faced a number of challenges. First, facility needs required the availability of sufficient cold storage for the produce and a commercial kitchen, including a dishwasher. Second, unpredictable weather and the unexpected intrusion of elk and deer altered the weekly availability of produce. This required recipe flexibility and anticipation of alternatives. Finally, logistics involving produce acquisition from six farms and transport to site-specific cold storage a day or two prior to each session was time intensive. One person was tasked to communicate each week with all participating farms as to the type and amount of produce available; calculating which produce items should be acquired from each farm; and scheduling produce pick-up. It is important to note the limitations of the small convenience sample, which limited subgroup analysis and generalizability; the lack of a control or comparison group; and self-reported data and measurement of dietary intake. Additional limitations included resource demands, lack of a process evaluation, and lack of data on non-completers. Nevertheless, we collected baseline and post-program data and enhanced the survey data with children’s and parents’ focus groups. There were a number of strengths to CwS4H: community support in providing access to appropriate program sites; the creation of a common price list for all farms that is now used in other community programs; the active engagement of parent–child pairs; an abundance of produce varieties in three growing seasons; the enhancement of opportunities for children to take ownership of produce selection for in-session preparation and take-home GFBs; and the reciprocal exchange of information, recipes, and strategies with the *¡Haz Espacio para Papi!* (HEPP, Make Room for Daddy!) program for Mexican-heritage families in border communities in South Texas [56].

## 5. Conclusions

This small study demonstrated that a community-based program that actively engaged parent–child pairs and focused on local vegetables produced during three distinct growing seasons could lead to positive experiences and outcomes. Programs that seek to improve children’s acceptability and consumption of vegetables should focus on the involvement of children in cooking activities [24].

## Figures and Tables

**Figure 1 nutrients-15-04851-f001:**
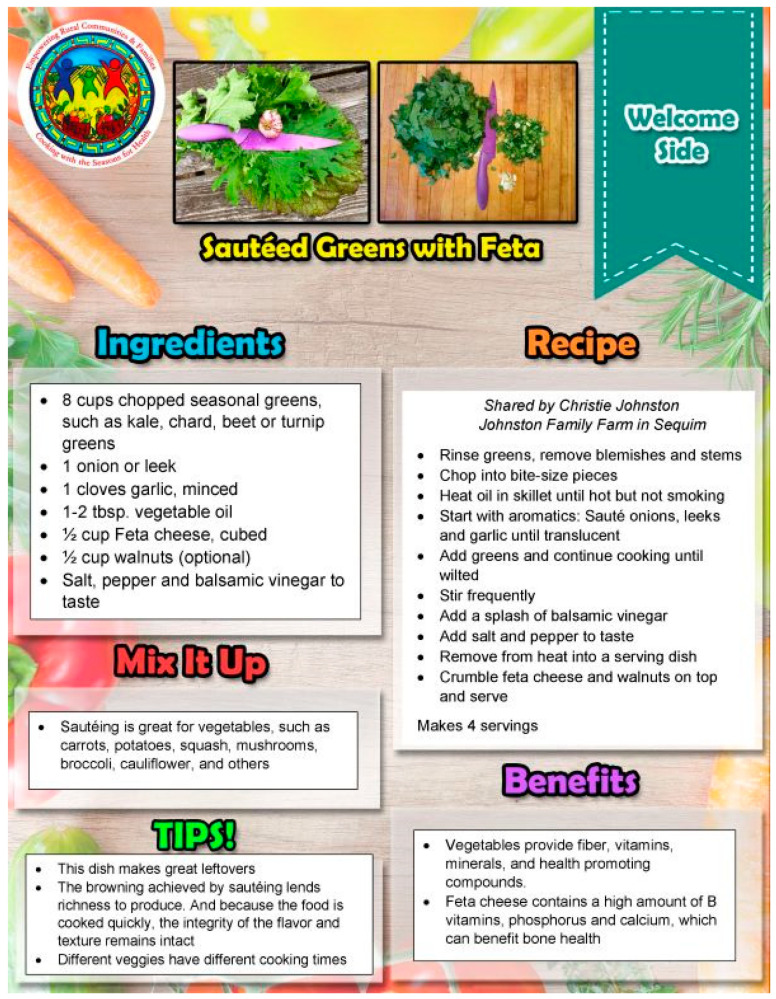
Example of session tasting.

**Figure 2 nutrients-15-04851-f002:**
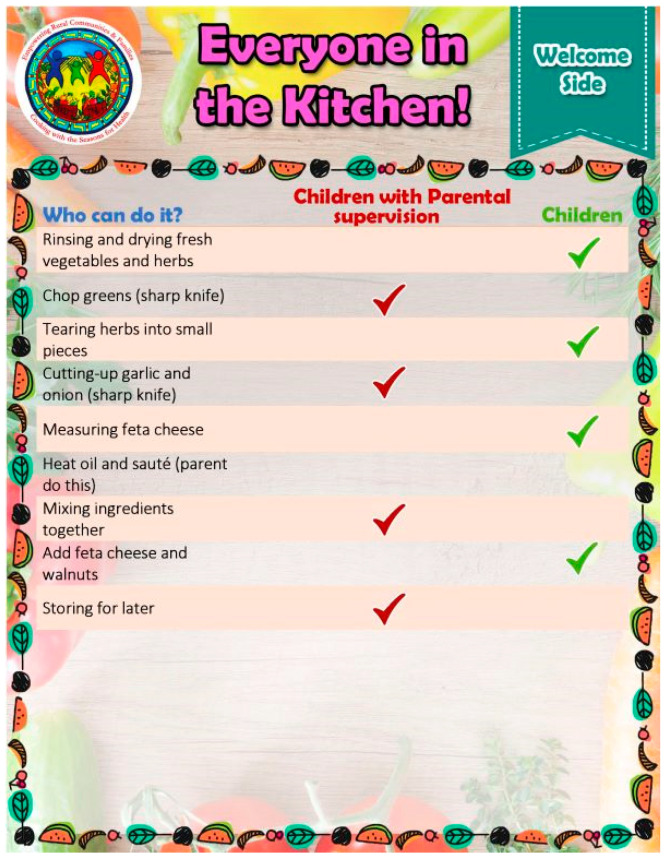
Tasting—Everyone in the Kitchen.

**Figure 3 nutrients-15-04851-f003:**
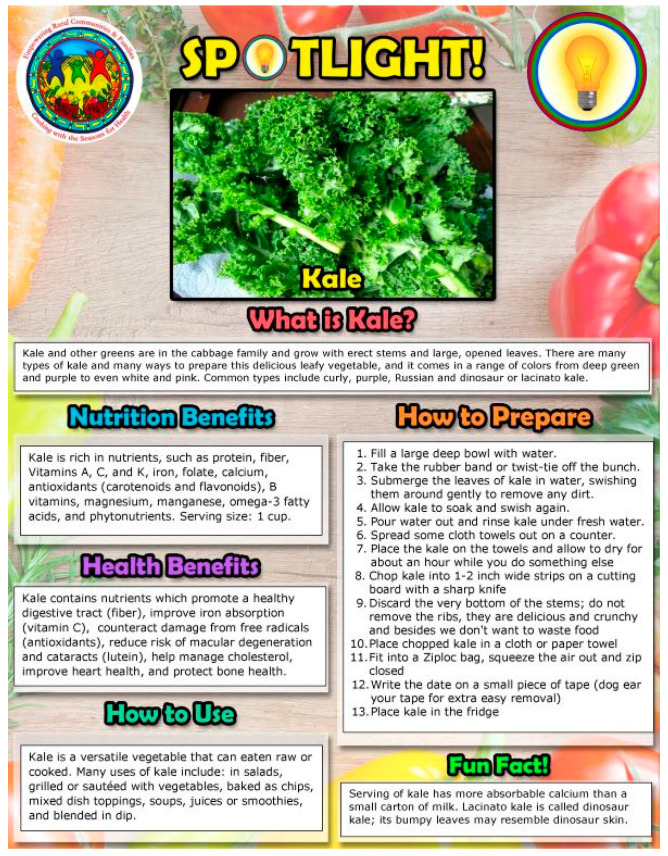
Spotlight on kale.

**Figure 4 nutrients-15-04851-f004:**
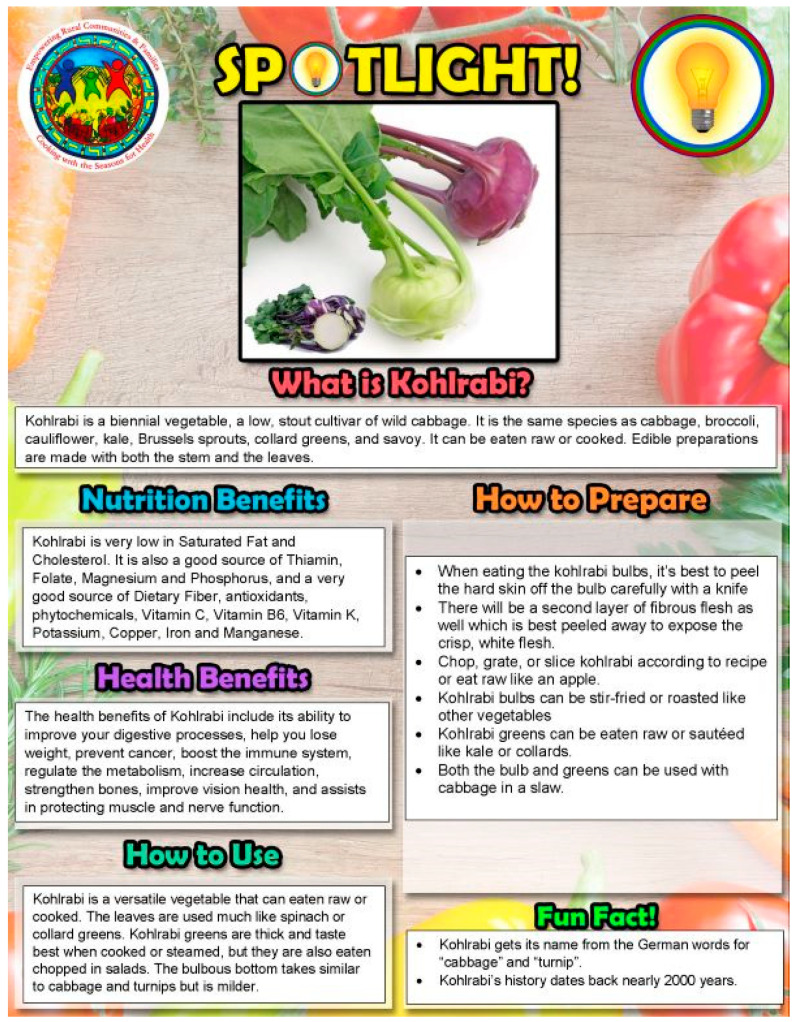
Spotlight on kohlrabi.

**Figure 5 nutrients-15-04851-f005:**
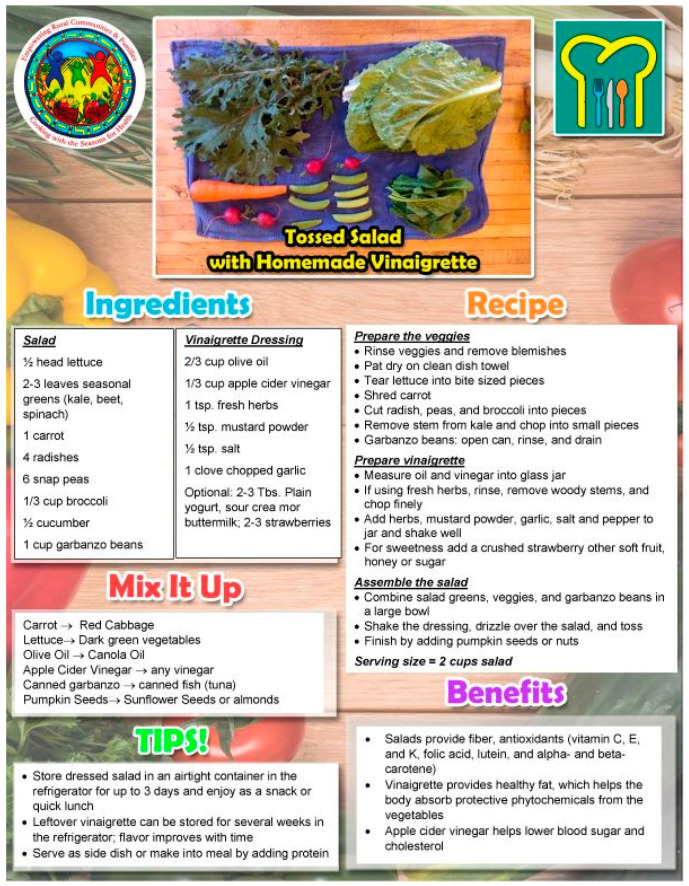
Session 1 recipe.

**Figure 6 nutrients-15-04851-f006:**
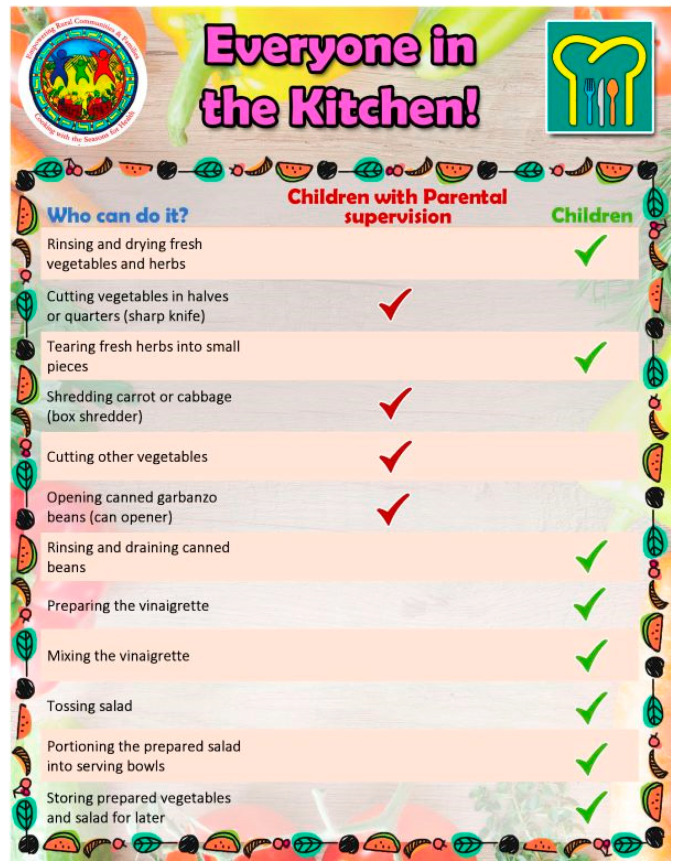
Session 1 recipe—Everyone in the Kitchen.

**Table 1 nutrients-15-04851-t001:** Session themes.

Session	Theme
1	Healthy Foundations—It Starts at Home (Surprising Veggies)
2	Healthy Families Cook and Eat Together (Veggie Math)
3	Fun in the Kitchen! (Veggie Subgroups)
4	Nutrition Basics (Vary Your Veggies)
5	Healthy Choices for Home and on the Go
6	Healthy Families Going Forward

**Table 2 nutrients-15-04851-t002:** Example of week 1 “Healthy Foundations” with activities and estimated time.

Component	Session 1 Activity	Estimated Time
Tasting recipe lesson	Three recipes: Cucumber slushy Sautéed greens with feta Kale ranch dip with veggies	As parent–child pairs arrived
Introduction	Greetings and introductions	10 min
Interactive lesson	Healthy meal Vegetable identification Hygiene and food safety Handwashing competition Food safety	20 min
Spotlight vegetable	Kale Garbanzo beans	10 min
Cooking lesson and hands-on meal preparation	Recipe demonstration How to wash and store vegetables Recipe: tossed salad with homemade vinaigrette	45 min
Eating together and recap	MyPlate Plan for next week Today’s session	20 min
Good Food Bag	Prepare and distribute	15 min

**Table 3 nutrients-15-04851-t003:** Session spotlights, tasting recipes, main recipes, nutrition education, and skills building.

Session	Spotlight	Testing Recipes and Minilesson	Main Recipe	Nutrition Education—Key Messages	Skills Building
1	Kale Garbanzo beans	Cucumber slushy. Sautéed greens with feta. Kale green ranch dip with veggies. Mini-lesson: Introduce MyPlate, healthy eating basics, vegetable identification, cooking with more vegetables.	Tossed salad with homemade vinaigrette.	Cooking together with more vegetables. Family favorites in a healthier way. Vegetables are a quick and easy snack. Benefits of seasonal greens.	Demonstrate awareness while working in the kitchen. Apply food safety/hygiene and food-handling practices. Wash and prepare a tossed salad. Measuring skills.
2	Broccoli Beets	Monster drink (kale and fruit). Yummy peanut butter/yogurt dip with veggies. Mini-lesson: Health benefits of vegetables and fruits, review MyPlate.	Grilled zucchini.	Children will eat more vegetables if involved in the preparation. Cooking together for a family meal as a family tradition. Cooking healthier staple meals.	Knife skills. Cut vegetables into strips. Grilling vegetables. Select healthier oils. “eye-balling” measures.
3	CauliflowerSummer squash	Garlicky white bean dip with celery and carrot sticks. Cucumber slushy. Veggie pizza. Mini-lesson: Composition of a healthy meal, planning a family meal, vegetable categorization activity.	Roasted seasonal vegetables.	Children can help safely prepare vegetables. Different vegetables contain different important nutrients. Beans are a great source of fiber and protein. Reminders that MyPlate offers balance and variety.	Washing different types of vegetables. Slice, dice, cube vegetables. Store vegetables. Safely using oven to roast vegetables. Using blender and marinating.
4	CarrotsKohlrabi	Fruit shake. Cauliflower and broccoli salad. Dry-roasted garbanzo beans. Minilesson: Reading nutrition labels, working with canned and frozen vegetables, vegetable vitamin A and C content activity.	Vegetable pizza.	Children taking the lead in food preparation. Include strange-looking vegetables. Importance of vitamins A and C for health. Children can help read nutrition labels.	Simmering (ability to simmer and why sauces are reduced with simmering). Preparing a baking sheet. Safely use an oven. Wash and prepare mushrooms and bell pepper. Use box grater.
5	Cabbage	Warm apple cider. Winter sunshine (roasted carrots). Crispy kale chips. Minilesson: Nutrients found in a variety of colorful vegetables, tasting activity, list different ways to cook vegetables activity.	Mommy (and daddy) ramen.	Plan and shop for groceries as a family activity. Children taking the lead encourages the family to eat more nutritious foods. Plan family meals with MyPlate recommendations.	Peeling. Sautéing. Wash and chop vegetables. Boiling. Chop into bite-size pieces.
6	Brussels sprouts	Curried squash soup. Pumpkin bread. Braised brussels sprouts. Minilesson: Making less expensive items healthier with vegetables, cook with “what you got”, recap.	Beet soup (borscht).	Growing children need more vegetables. Soups are perfect for fall and winter. Encourage children to take the lead in cooking.	Use immersion blender. Chop and peel vegetables. Reinforce previous skills. Modify recipes.

**Table 4 nutrients-15-04851-t004:** Good Food Bag produce for sessions 2, 4, and 6.

Session 2		Session 4		Session 6	
Produce Item	Amount	Produce Item	Amount	Produce Item	Amount
Zucchini	1 lb. (~3)	Kohlrabi	1 each	Garlic	1 bulb
Kale	1 bunch	Cauliflower	1 head	Tomatoes	1 lb. (~6)
Carrots with tops	1 lb. (~8)	Cabbage	1 head	Cherry tomatoes	1 pint
Snap/snow pea mix	1 bag	Parsley	1 bunch	Braising mix	1 bag
Raspberries	½ pint	Carrots	1 lb. (~8)	Pie pumpkins	1 each
Lettuce	1 head	Cucumbers	1 lb. (~6)	Savoy cabbage	1 head
Beets with tops	1 lb. (~3)	Broccoli	2 heads	Cranberry potatoes	3 lbs.
Cauliflower	1 head	Tomatoes	1 lb. (~6)	Ozette potatoes	3 lbs.
Garlic	1 bulb	Celery	1 bunch	Spaghetti squash	1 each
Parsley	1 bunch	Leeks	1 lb. (~4)	Butternut squash	1 each
		Basil	1 bunch	Long pie pumpkins	1 each
		Onions	~1 lb.	Sweet potato squash	2 each
				Collards	1 bunch
				Beets	1 bunch
				Romaine lettuce	1 head

**Table 5 nutrients-15-04851-t005:** *Cooking with the Seasons for Health* outcome variables and food preparation and cooking questions.

Variable	Example Item	# Items	Response Options	Pre-Program	Post-Program
**Child survey**					
Nutrition knowledge	What information can we learn from MyPlate?	6	0–1 (incorrect response = 0, correct response = 1)	X	X
Vegetable preference/liking	What do you think of the following vegetables? (17 vegetables)	17	0–1 (dislike or never tried = 0, favorite or like = 1)	X	X
Self-efficacy for food preparation and cooking	What is your level of confidence with certain food preparation and cooking activities?	15	0–1 (unable to do or need help = 0, can do on own = 1)	X	X
Parent modeling	When you are with your parent(s), how often do they eat vegetables at dinner?	4	0–1 (sometimes or hardly ever or never = 0, usually/always = 1)	X	X
Vegetable intake	On a normal day, how many times a day does this child eat vegetables?	1	0–2 (none = 0, 1–2 times = 1, 3 or more times = 2)	X	X
Willing to try new foods	You eat food that you have never eaten before.	9	0–1 (never or sometimes = 0, usually or always = 1)	X	
Attitude for cooking	How do you feel about making snacks with vegetables?	4	0–1 (don’t or really don’t like=0, really or kind of like = 1)	X	
Assist parents	How often do you help prepare dinner?	2	0–1 (never or once in a while = 0, weekly = 1)	X	
Confidence in preparing/eating vegetables	What is your level of confidence? Make vegetable snacks or foods for yourself.	6	0–1 (unsure = 0, moderately or very sure I can = 1)	X	
Vegetables tried last month	Have you tried in the past month? (23 vegetables)	23	0–1 (no = 0, yes = 1)		X
Food activities performed	In the past, … ? You ate a food that you have never had before.	13	0–1 (no = 0, yes = 1)		X
**Parent survey**					
Nutrition knowledge	What information can we learn from MyPlate?	6	0–1 (incorrect response = 0, correct response = 1)	X	X
Nutrition behavior	Do you eat food past its ‘use by’ date?	5	0–1 (sometimes or always = 0, never = 1)	X	X
Vegetable preference/liking	What do you think of the following vegetables? (list of 16 vegetables)	16	0–1 (dislike or never tried = 0, favorite or like = 1)	X	X
Attitude for food preparation activities	When preparing food, you are confident that you can deal with unexpected results.	13	0–1 (disagree = 0, moderately or strongly agree = 1)	X	X
Positive attitude for cooking	You find cooking a very fulfilling activity.	10	0–1 (disagree = 0, moderately or strongly agree = 1)	X	X
Time as cooking barrier	You wish you had more time to plan meals.	5	0–1 (disagree = 0, moderately or strongly agree = 1)	X	X
Family food practices	How often during a typical week do you talk about eating healthy foods with your child?	5	0–1 (never or less than weekly = 0, at least weekly = 1)	X	X
Confidence in preparing vegetables	What is your confidence in preparing …?	20	0–1 (unsure I can = 0, moderately or very sure I can = 1)	X	X
Vegetable intake	What was the number of times vegetables were eaten yesterday?	1	0–2 (none = 0, 1–2 times = 1, 3 or more times = 2)	X	X
Family support for vegetables	Who would support you in making it easier for your child to eat vegetables?	4	Open ended	X	X
Confidence in performing food activities	How confident are you …? Making vegetable-focused side dishes at home.	12	0–1 (not confident = 0, moderately or extremely confident = 1)		X
Vegetable availability	How often is the following true? We have vegetables in my home	5	0–1 (sometimes or not at all = 0, almost every day = 1)		X
Child participation	Does you child do the following on his/her own? Make snack with vegetables.	11	0–1 (no = 0, yes = 1)		X

**Table 6 nutrients-15-04851-t006:** Parent and child characteristics from *Cooking with the Seasons for Health* surveys.

		Pre-Program (*n* = 30 Parent–Child Pairs)			Post-Program (*n* = 23 Parent–Child Pairs) *		
		*n* (%)	Mean ± SD	Range	*n* (%)	Mean ± SD	Range
Town							
	Port Angeles	16 (53.3)			10 (43.5)		
	Sequim	14 (46.7)			13 (56.5)		
Parent							
	Dad	4 (13.3)			4 (17.4)		
	Mom	26 (86.7)			19 (82.6)		
Child							
	Boy	17 (56.7)			14 (60.9)		
	Girl	13 (43.3)			9 (39.1)		
Child age							
	8 y	8 (26.7)			7 (30.4)		
	9 y	8 (26.7)			6 (26.1)		
	10 y	7 (23.3)			4 (17.4)		
	11 y	7 (23.3)			6 (26.1)		
Parent age (y)			38.7 ± 8.6	26–71		40.3 ± 9.0	26–71
Parent education							
	Some college	18 (60)			13 (56.5)		
Parent marital status							
	Married/living with partner	21 (70)			19 (82.6)		
Household composition							
	Adults		2.0 ± 0.7	1–4		2.1 ± 0.7	1–4
	Children		2.7 ± 1.2	1–7		2.9 ± 1.3	1–7
	Total		4.6 ± 1.5	2–10		5.0 ± 1.5	3–10

* A total of 23 parent–child pairs completed the program; 23 parents and 22 children completed the post-program survey.

**Table 7 nutrients-15-04851-t007:** Participation in nutrition assistance programs.

	Current (Summer)		School Year	
	*n*	%	*n*	%
SNAP	11	36.7	11	36.7
Free/reduced breakfast	0	0	15	50
Summer meals program	14	46.7	0	0
WIC	5	16.7	6	20
Free/reduced lunch	3	13	15	50
Food bank/food pantry	7	23.3	8	26.7
None	9	26.7	8	26.7

**Table 8 nutrients-15-04851-t008:** Assessment of *Cooking with the Seasons for Health* Children Participants (*n* = 22).

	Pre-Program			Post-Program					
Measurement	Mean	SD	Range	Mean	SD	Range	Mean Difference	95% CI of Difference	Significance
Nutrition knowledge	3.77	1.44	1–6	4.36	1.25	2–6	−0.59	−1.06, −0.12	*p* = 0.016
Vegetable preference	9.45	3.13	4–14	10.27	4.03	3–16	0.69	−2.25, 0.61	*p* = 0.248
Vegetables never tried/not sure	1.82	1.84	0–8	0.95	1.33	0–6	0.86	0.19, 1.54	*p* = 0.015
Self-efficacy for food preparation and cooking	8.00	2.41	4–13	10.14	2.42	7–15	−2.14	−3.44, −0.84	*p* = 0.002
Parental modeling	3.54	0.80	1–4	3.64	0.73	1–4	−0.09	−0.50, 0.32	*p* = 0.648
Willing to try new food	4.5	1.63	1–7						
Attitude for cooking	2.41	1.33	0–4						
Confidence in preparing/eating vegetables	2.04	1.91	0–6						
Vegetables tried in past month				13.27	3.76	7–21			
Food activities perform				9.18	2.30	5–13			

**Table 9 nutrients-15-04851-t009:** Assessment of *Cooking with the Seasons for Health* Parent Participants (*n* = 23).

	Pre-Program			Post-Program					
Measurement	Mean	SD	Range	Mean	SD	Range	Mean Difference	95% CI of Difference	Significance
Nutrition knowledge	5.04	0.92	3–6	5.69	0.63	4–6	−0.65	−1.03, −0.27	*p* = 0.002
Nutrition behavior	2.30	1.22	0–4	2.61	1.47	0–5	−0.30	−0.83, 0.22	*p* = 0.245
Vegetable preference	13.78	3.68	0–16	14	1.78	8–16	0.71	13.23, 14.77	*p* = 0.763
Attitude for food preparation activities	8.69	3.05	4–13	10.74	2.47	2–13	−2.04	−3.21, −0.88	*p* = 0.001
Positive attitude for cooking	4.78	3.04	0–10	5.35	2.90	0–10	−0.56	−1.64, 0.51	*p* = 0.287
Time as cooking barrier	1.83	1.58	0–5	1.56	1.70	0–4	0.32	−0.39, −0.92	*p* = 0.417
Family food practices	3.30	1.02	1–5	3.83	0.89	3–5	−0.52	−1.01, −0.04	*p* = 0.036
Confidence in preparing vegetables	16.74	3.62	8–20	17.26	3.58	6–20	−0.52	−1.92, 0.88	*p* = 0.447
Confidence food activities perform				9.48	3.15	1–12			
Vegetables tried in past month				13.27	3.76	7–21			
Food security				1.52	2.02	0–7			

Note: Statistical significance with Bonferroni correction at *p* < 0.006.

## Data Availability

Survey and focus group data presented in this study are not available.

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
