# Peer review of "Cooking with the Seasons for Health (CwS4H): An Innovative Intervention That Links Nutrition Education, Cooking Skills, and Locally Grown Produce to Increase Vegetable Intake among Limited-Resource Parent–Child Dyads in Rural Washington"

_nutrients, 2023, doi:10.3390/nu15224851_

Round 1

Reviewer 1 Report

Comments and Suggestions for Authors

Methods - theoretical foundation - clarify how the study methods are based in/linked to each of the 3 theories mentioned.

Consent - Since this study involves children give some detail of the consent process for adults and children.

Data analysis - explain the Sort and Sift, Think and Shift qualitative data analysis approach.

Sample characteristics - given the fair size difference in program completion at the 2 sites, was there a comparison of possible differences between the 2 groups, or difference between program completers and non-completers, or some other observations to explain the different rates of completion?

Table 8 - cannot see full table

Results - Child and parent focus group  and farmer interview data could be better organized into distinct themes with names.

Discussion - starts off with too much detail on information repeated from previous sections.  Only need a brief recap of main findings.

The discussion finds many instances where the current study results are similar to other studies.  Make it more clear how this study has built on or enhanced past findings. Emphasize unique findings. What will the findings mean for the children moving forward in their current context of rural living, with many of them being food insecure to some degree?

Comments on the Quality of English Language

minor editing required

Reviewer 2 Report

Comments and Suggestions for Authors

This study presents the development and evaluation of "Cooking with the Seasons for Health (CwS4H)," a collaborative initiative between an academic institution and a rural food bank in Washington, aimed at promoting fresh produce consumption among parent-child dyads. Through hands-on strategies across six sessions during three distinct growing seasons, the program effectively enhanced the knowledge, attitudes, and skills related to vegetable consumption. The integration of local farmers, interactive education, and practical food preparation highlights a comprehensive approach to addressing nutritional behaviors in rural communities.

Abstract:

The results are concise. However, it might be worth noting that using percentages or other form of quantification might make the impact more evident.

The abstract uses various terms such as “nutrition behaviors”, “food preparation activities”, “food-related activities”. It might be helpful to streamline this terminology.

Introduction:

Page 1:

First sentence: The term” limited-resource children” could be rephrased as “children from limited-resource families” for clarity.

A brief definition of the “Healthy Eating Index” could help readers unfamiliar with the term

Page 2, paragraph 1:

Consider starting with a broader statement about the complexity of children’s eating habits before delving into details.

The sentence “ Acquisition of cooking skills at …later in life” seem to reiterate the point made in the previous sentence. Consider combining the two statements.

Page 2, paragraph 3:

The detailed description of programs like “Fun with Food”, “Cooking Matters for Families”, consider summarizing the main outcomes or impacts to avoid overloading the reader with details.

Page 3 paragraph 1:

Consider combining this sentence with the aim paragraph. It sets a clear problem statement that the rest of the aims to address.

Materials & Methods

It would be insightful to know if there were any other feedback mechanism such as mid-program evaluations or feedback in addition to pre- and post- program surveys.

Data analysis:

Provide more details regarding within-person change, which specific statistical test? Consider adjusting for multiple comparison to reduce the risk of Type 1 error.

The “Sort and Sift, Think and Shift” approach is mentioned for qualitative data analysis. Consider providing a brief description or rationale for its use.

Discussion:

Consider adding more limitations such as lacking process evaluation, short-term evaluation.

Comments on the Quality of English Language

No issues 

Round 2

Reviewer 1 Report

Comments and Suggestions for Authors

My comments have been addressed.

Author Response

Comments from reviewer 1 has been addressed. Nothing remains to address.

Reviewer 2 Report

Comments and Suggestions for Authors

Data Analysis:  Because you have a lot of paired t tests, it would then be appropriate to make the adjustment for multiple testing. For example, Bonferroni is a simple procedure, if you want to test at α=0.05 p=0.05 for 10 paired t test, you should consider tests to show significant differences from zero if their p-values are ≤0.05/10=0.005

add a limitation about the lack of process evaluation. 
